# Histone H4 potentiates neutrophil inflammatory responses to influenza A virus: Down-modulation by H4 binding to C-reactive protein and Surfactant protein D

I-Ni Hsieh[1], Mitchell White[1], Marloes Hoeksema[2], Xavier Deluna[1], Kevan Hartshorn[1]*

1 Department of Medicine, Section of Hematology Oncology, Boston University School of Medicine, Boston, Massachusetts, United States of America, 2 University of Amsterdam, Amsterdam, Netherlands

* khartsho@bu.edu

**Data Availability Statement:** All relevant data are within the manuscript and its Supporting information files.

## Abstract

Neutrophils participate in the early phase of the innate response to uncomplicated influenza A virus (IAV) infection but also are a major component in later stages of severe IAV or COVID 19 infection where neutrophil extracellular traps (NETs) and associated cell free histones are highly pro-inflammatory. It is likely that IAV interacts with histones during infection. We show that histone H4 binds to IAV and aggregates viral particles. In addition, histone H4 markedly potentiates IAV induced neutrophil respiratory burst responses. Prior studies have shown reactive oxidants to be detrimental during severe IAV infection. C reactive protein (CRP) and surfactant protein D (SP-D) rise during IAV infection. We now show that both of these innate immune proteins bind to histone H4 and significantly down regulate respiratory burst and other responses to histone H4. Isolated constructs composed only of the neck and carbohydrate recognition domain of SP-D also bind to histone H4 and partially limit neutrophil responses to it. These studies indicate that complexes formed of histones and IAV are a potent neutrophil activating stimulus. This finding could account for excess inflammation during IAV or other severe viral infections. The ability of CRP and SP-D to bind to histone H4 may be part of a protective response against excessive inflammation in vivo.

## Introduction

Influenza A viruses continue to pose a major threat to human health. While most people recovery from IAV infection without major complications, some subjects suffer severe morbidity and mortality due to viral pneumonia, bacterial superinfection or exacerbation of underlying cardiac disease. Of major concern has been the possible emergence of radically new IAV strains from avian or porcine sources causing a pandemic. The mortality rate from human infection with avian IAV is very high and prior pandemics have varied in severity with the 1918 pandemic carrying the highest mortality rate. Currently we are in the grips of the novel SARs-COV2 pandemic which bears some similarities to prior IAV pandemics but with a mortality rate exceeding that of seasonal IAV or even of most IAV pandemics excluding that

**Funding:** This work was supported by NHLBI R01HL069031 and funders had no role in study design, data collection and analysis, decision to publish, or preparation of the manuscript.

of 1918. While avian or pandemic IAV and SARS-CoV2 will likely be found to differ in various respects, they both involve profound inflammation which has been termed cytokine storm and destruction of the delicate alveolar capillary interface in the lung. In both COVID19 and IAV infection there is a profound elevation of acute phase reactants like CRP and Serum amyloid A [1, 2]. The role of these acute phase reactants in host defense (or host injury) during viral pneumonia is not clear. Another common feature emerging in recent literature is a role for neutrophils and neutrophil extracellular traps (NETs) during both IAV and COVID 19 [3–6]. Neutrophils can play beneficial or harmful roles in IAV infection but in the context of severe viral pneumonia or other severe lung injuries (e.g. ARDS) there is concern that they have a damaging role. Histones are major component of NETs that have potent effects when in the extracellular space.

Free histones resemble cationic antimicrobial peptides and we previously showed the arginine rich histones H3 and H4 have antiviral activity against IAV [7]. However, many studies now indicate that extracellular histones are predominantly harmful in vivo through profoundly inducing inflammation and activating coagulation. Histones have been shown to contribute to the pathophysiology of sepsis and various forms of lung injury (including IAV) [6, 8–13]. In these settings, antibodies to histones have been shown to be protective [12, 14]. Histones can be released from damaged or dying cells or through neutrophil extracellular trap (NET) formation in the case of neutrophils [15]. Extracellular histones act like other damage-associated molecular patterns (DAMPs) [16] and they can bind to platelets and induce profound thrombocytopenia in mice [17, 18] and promote thrombin generation and hypercoagulation [19]. Since NETs and histones have been implication in pathogenesis of IAV and likely interact with the virus during severe infection, and IAV induces NET formation in vitro, we studied interaction of histones and IAV with human neutrophils. We recently reported that histone H4 directly activates neutrophils through causing membrane permeabilization leading to sustained calcium influx, respiratory burst activation, degranulation and generation of IL-8 [20].

C-reactive protein (CRP) has been reported to attenuate the endothelial cell damage, vascular permeability, and platelet aggregation caused by high levels of extracellular histones in vitro and in vivo [21]. We therefore tested how CRP modulates effects of histones on neutrophils. Surfactant protein D (SP-D) is a host defense lectin shown to be protective against IAV [22]. It has direct antiviral activity but also reduces lung inflammation. Levels of SP-D rise during IAV infection. In addition it has been shown to bind to the spike protein of SARS-CoV1 [23] and to become elevated in the serum during SARS-CoV1 infection [24]. SP-D has been shown to bind to NETs [25]. We, therefore, also tested how SP-D modulates interactions of histones with neutrophils.

In this paper, we first evaluate in greater depth the interactions of histone H4 with IAV and then how it modulates interactions of IAV with neutrophils. We focused on predominantly on human histone H4 in part because we found, as did Abrams et al [21], that other purified histone preparations available commercially contained endotoxin. In addition, histone H4 has been implicated in several studies of histone induced inflammation [12, 26]. We confirm that IAV binds to NETs containing histones and that histone H4 binds to various strains of IAV. We found that histone H4 aggregates IAV particles and markedly potentiates neutrophil respiratory burst responses to the virus. While CRP did not inhibit IAV infection, cause viral aggregation or alter viral interactions with neutrophils in its own right, it binds to histone H4 and markedly reduced its interactions with the virus and its effects on neutrophils. Surfactant protein D also bound to histone H4 and reduced its effects on neutrophils. For experiments with SP-D we also tested effects of recombinant constructs consisting of just the neck and carbohydrate domain (NCRD) of the molecule to show that this domain mediates binding to H4.

These findings may provide some insight into the roles of CRP and SP-D during severe viral infection.

## Methods

### Ethics statement

Blood collection for isolation of neutrophils was done with informed consent as approved by the Institutional Review Board of Boston University School of Medicine. The Institutional Review Board specifically approved this study and also approved the consent form for the study. Approval for the study is renewed yearly in June. The blood donors were healthy volunteers and they all signed the written consent form prior to each donation.

### Proteins and reagents

Recombinant histone H4 was purchased from New England Biolabs (Ipswich, MA, USA). PMA and fMLP were purchased from Sigma-Aldrich (St. Louis, MO, USA). Recombinant Human C reactive protein (CRP) and anti-C reactive protein antibody were purchased from Abcam (Cambridge, United Kingdom). Recombinant human SP-D, NCRD and D325A +R343V mutant NCRD were kindly provided by Dr. Erika Crouch. Recombinant human SP-D were produced in CHO cells as previously described [27]. There recombinant SP-D can can contain trimers, dodecamers and high molecular weight multimers and we used the isolated dodecamers for consistency. Trimeric NCRD fusion proteins consist of the neck domain and carbohydrate recognition domains of SP-D combined with NH2-terminal tags that contain a His-tag and S-protein binding site that permit purification and/or detection [28]. D325A+R343V was produced and characterized as previously described [29, 30]. All showed a single major band of appropriate size by SDS-PAGE and demonstrated the expected decrease in mobility on reduction, consistent with appropriate formation of intrachain disulfide bonds. The endotoxin level of all SP-D preparations was 0.1 to 0.5 EU/mL (Limulus Lysate Assay; Cambrex, Walkersville, MD). Antibody against SP-D (246–08) was kindly provided by Dr. Uffe Holmskov. It was raised by inoculating mice with 10 μg/ml human SP-D as previously described [31]. Histone H4 was obtained from New England Biolabs and was shown to have less than 0.016 EU per ml of endotoxin.

### Virus preparations

Philippines 82/H3N2 (Phil82) strain was kindly provided by Dr. E. Margot Anders (Univ. of Melbourne, Melbourne, Australia). The PR-8 (1934 H1N1) strain was provided by Jon Abramson (Wake Forest University, Winston-Salem, NC). These IAV strains were grown in the chorioallantoic fluid of ten day old chicken eggs and purified on a discontinuous sucrose gradient as previously described [32]. The viruses were dialyzed against PBS to remove sucrose, aliquoted and stored at -80˚C until needed. Post thawing the viral stocks contained ~5x10$^8$ infectious focus forming units/mL. The number of virus particles per mL was calculated using the infectious focus assay and this was used to calculate multiplicity of infection (MOI) for the various studies involving neutrophils or epithelial cells. California 2009 H1N1 strain was derived by reverse genetics and grown in Madin-Darby canine kidney (MDCK) cells. For binding of IAV to histone H4, IAV particles were labeled with Alexa Fluor-594 using Alexa Fluor 594 Protein Labeling Kit (Invitrogen, Carlsbad, CA). Concentrated virus stock was incubated with the Alexa Fluor in sodium bicarbonate buffer (pH 8.3) for one hour at room temperature. The preparation was then dialyzed overnight against PBS at 4˚C.

## Confocal microscopy

Neutrophils were resuspended in PBS supplemented with $Ca^{2+}$ and $Mg^{2+}$ and allowed to adhere on Poly-L-Lysine coated glass bottom culture dishes (MatTek corporation, Ashland, MA) for 1hr in a $CO_2$ incubator. After 1hr, unadhered cells were removed and adhered cells were incubated for 3 hours in a $CO_2$ incubator with Fluorescein isothiocyanate (FITC)-labeled IAV (Phil82 strain). Dishes were then washed with PBS and fixed with 1% paraformaldehyde. DAPI 350 was used to stain the DNA, and alexa-conjugated anti-histone H4 antibody was used to stain histone H4. Confocal pictures were taken at Zeiss LSM510 (LSEB) on 100x resolution.

## Human neutrophil preparation

Neutrophils from healthy volunteers were isolated to > 95% purity by using dextran precipitation, followed by Ficoll-Paque gradient separation for the separation of mononuclear cells (layering above the Ficoll-Paque) and neutrophils (below the Ficoll-Paque). The neutrophils were purified further by hypotonic lysis to eliminate any contaminating erythrocytes, as previously described [32]. Cell viability was determined to be >98% by trypan blue staining. The isolated neutrophils were resuspended at the appropriate concentrations in control buffer (PBS) and used within 2 hours. All incubations of virus or proteins with cells were carried out in PBS with magnesium and calcium unless otherwise indicated. All of these incubations were done at 37˚C in a humidified incubator with 5% $CO_2$.

## Binding of IAV to histone H4

96-well black clear-bottom plates (Scientific, Pittsburgh, PA) were coated overnight with 10 μg/mL (890 nM) H4 and 15 mg/mL (890 nM) BSA (fraction V, fatty-acid-free and low endotoxin; SigmaAldrich, St. Louis, MO) in PBS. For blocking, 100 μL of superblock (1:2 in $H_2O$) (Pierce, Rockford, IL) was added to wells and immediately dumped. Wells were then incubated with 100 μL superblock (1:2 in $H_2O$) for 5 minutes. 50 μL of sample containing diluted Alexa Fluor 594-labeled Phil82/PR8/Cal09 was added to each well for 45 minutes. Fluorescence was measured using a POLARstar OPTIMA fluorescent plate reader (BMG Labtek, Durham, NC).

## Measurement of viral aggregation by light transmission

Viral aggregation was measured by assessing light absorbance by stirred suspensions of IAV at 350 nM using a Perkin Elmer Lambda 35 ultraviolet/Vis spectrophotometer.

## Fluorescent focus assay of IAV infectivity

MDCK cell monolayers were prepared in 96 well plates and grown to confluency. These cells were purchased from the American Type Culture Collection (Manassas, VA) and propagated in the undifferentiated state in standard tissue culture flasks. These layers were then infected with diluted IAV preparations for 45 min. at 37˚C in PBS. MDCK cells were tested for presence of IAV infected cells after 18 hours of virus addition using a monoclonal antibody directed against the influenza A viral nucleoprotein (provided by Dr. Nancy Cox, CDC, Atlanta, GA) as previously described. Before adding to cell layers, IAV was pre-incubated for 30 min. at 37˚C with PBS, histone H4, CRP, SP-D or the combination of H4 and CRP/SP-D. The multiplicity of infection (MOI) was approximately 0.1. MOI was calculated based on number of cells at confluence. After 45 minutes, the plate was washed, followed by 18 hours incubation at 37˚C in tissue culture media (media used was specific for the cell line used in the

experiment and as per manufacturer's instructions). After 18 hours, MDCK cells were washed with PBS and fixed with chilled 80% acetone for 10 minutes. Presence of IAV infected cells was detected using a primary mouse monoclonal antibody (1::100 dilution) directed against the influenza A viral nucleoprotein (EMD Millipore, MA) as previously described [33]. A rhodamine labeled secondary (1:1000) antibody (EMD Millipore, MA) was used to detect primary antibody. Fluorescent positive cells were counted visually on a fluorescent microscope (Nikon MVI, Avon, MA, US).

## Measurement of IAV uptake by neutrophils

Fluorescein isothiocyanate (FITC)-labeled IAV (Phil82 strain) was prepared and uptake of virus by neutrophils was measured by flow cytometry as described [33]. In brief, IAV was treated with various doses of histone H4 for 30 minutes at 37˚C. For experiments included CRP, histone H4 was pre-incubated with CRP for 30 minutes at 37˚C prior to the experiments. Then it was incubated with cells for 45 minutes at 37˚C in presence of control buffer. Trypan blue (0.2 mg/ml) was added to these samples to quench extracellular fluorescence. Following washing, the neutrophils were fixed with 1% paraformaldehyde and neutrophil associated fluorescence was measured using flow cytometry. The mean cell fluorescence (>2000 cells counted per sample) was measured.

## Measurement of neutrophil $H_2O_2$ production

$H_2O_2$ production was measured by assessing reduction in scopoletin fluorescence as previously described [34, 35]. In brief neutrophils were added to a mixture of scopoletin, sodium azide, and horseradish peroxidase, which were previously shown to maximize detection of $H_2O_2$. Results of this assay were previously correlated with results obtained by oxygen consumption, chemiluminescence and other assays [34–37]. Measurements were made using a POLARstar OPTIMA fluorescent plate reader (BMG Labtech, Durham NC). Virus alone or virus that had been pretreated with histone H4 for 30 minutes at 37˚C were added at Time 0 in the figure. For experiments with SP-D or NCRD, IAV was pre-incubated with these proteins for 30 minutes at 37˚C prior to addition to neutrophils.

## Measurement of intracellular calcium responses of neutrophils

Neutrophils were pre-loaded with Fura-2AM for 30 minutes at 37˚C. After washing, pre-loaded neutrophils at 2.5x10$^6$ cells/mL were added to the 96 well black plates (100 μL/well). Virus alone or virus that had been pretreated with histone H4 for 30 minutes at 37˚C was added at Time 135 in the figure. PBS control or fMLP were also added at Time 135 in the figure. For experiments included other peptides, histone H4 was pre-incubated with those peptides for 30 minutes at 37˚C prior to the experiments. Measurements were made using a POLARstar OPTIMA fluorescent plate reader (BMG Labtech, Durham, NC, USA).

## Measurement of neutrophil membrane depolarization

Neutrophils were incubated with Di-O-C53 and monitored on the fluorescent plate reader until fluorescence was stable and the histone H4, CRP, or the combination of both were added. In this assay a decrease in fluorescence indicates membrane permeabilization.

## Binding of histone H4 to CRP/SP-D/NCRD

Binding of histone H4 to CRP, SP-D and NCRD were assessed by solid-phase ELISA. 96-well plates were coated with 10 μg/mL histone H4 in coating buffer (15 mM $Na_2CO_3$, 35 mM

NaHCO$_3$, pH 9.6) overnight at 4˚C with PBS containing 2.5% (w/v) BSA (fraction V, fatty acid free, and low endotoxin, A8806; Sigma-Aldrich) as background control. After washing three times with PBS, the plates were blocked with PBS containing 2.5% BSA for 3 hours. These coated plates were then incubated with different concentrations of CRP, SP-D or NCRD and then washed with PBS containing 0.02% Tween 20, followed by addition of anti-CRP antibody against CRP, anti-SP-D antibody against SP-D or S protein-HRP conjugate (Novagen) against NCRD. Bound primary antibodies were detected with HRP-labeled goat anti-rabbit antibodies followed by incubation with tetramethylbenzidine (TMB) as a substrate (Bio-Rad). The reaction was stopped using 1 N sulfuric acid (Sigma). The optical density (OD) of the sample was measured on an ELISA plate reader at 450 nm wavelength. Background nonspecific binding was assessed by coating plates with fatty acid free BSA but no histone H4.

We also tested binding of histone H4 to CRP, SP-D or NCRD by co-precipitation and SDS-PAGE. Two μg histone H4 was incubated with or without CRP, SP-D or NCRD in PBS for 30 minutes at 37˚C. After incubation, sample was centrifuged at 1,200 x g for 5min. Supernatants and pellets were collected to fresh tubes and all samples were reduced and denatured by incubating in b-mercaptoethanol and boiling of samples for 5min at 95˚C. Samples were separated by SDS-PAGE on Mini-Protean Any kD TGX gel (BIO-RAD, Hercules, CA). For staining, the gel was incubated with GelCode Blue Stain Reagent (Pierce, Rockford, IL) for 1 hour.

## Measurement of the release of MPO

Human neutrophils were treated with indicated proteins for 2 hours. For experiments included two peptides, those peptides were pre-incubated together for 30 minutes at 37˚C prior to the experiments. Samples were then centrifuged for 5 minutes at 400 x g, and supernatants were collected. Exocytosis of myeloperoxidase (MPO) was measured with 3,3',5,5'-tetramethyl benzidine (TMB) as the substrate, and the reaction was stopped by adding 1N hydrochloric acid (Sigma). The optical density (OD) of the sample was read at 450 nm wavelength with POLARstar OPTIMA plate reader (BMG Labtech, Durham, NC, USA).

## Measurement of neutrophil caspase 3 activity

Human neutrophils were treated with indicated proteins for 5 hours. For experiments included two peptides, those peptides were pre-incubated together for 30 minutes at 37˚C prior to the experiments. Samples were then centrifuged for 5 minutes at 400 x g, and pellets were collected. Collected cells were then washed with PBS and lysed with lysis buffer. Caspase 3 activity of each sample was measured with EnzChek® Caspase-3 Assay Kit (Invitrogen™ Molecular Probes™). Ac-DEVD-CHO inhibitor was provided in the kit to ensure the observed signal is due to the activity of caspase-3-like proteases.

## Statistics

Statistical comparisons were made using Student's paired, two-tailed *t* test or ANOVA with post hoc test (Tukey's). ANOVA was used for multiple comparisons to a single control.

## Results

## I. Effects of histone H4 on IAV and neutrophil responses to IAV

**A. Histone H4 binds to and aggregates IAV and increases neutrophil uptake of the virus.** The confocal image shows that histone H4 was detected interacting with Phil82 IAV strain in IAV stimulated neutrophil extracellular traps (NETs) (Fig 1A). The image is

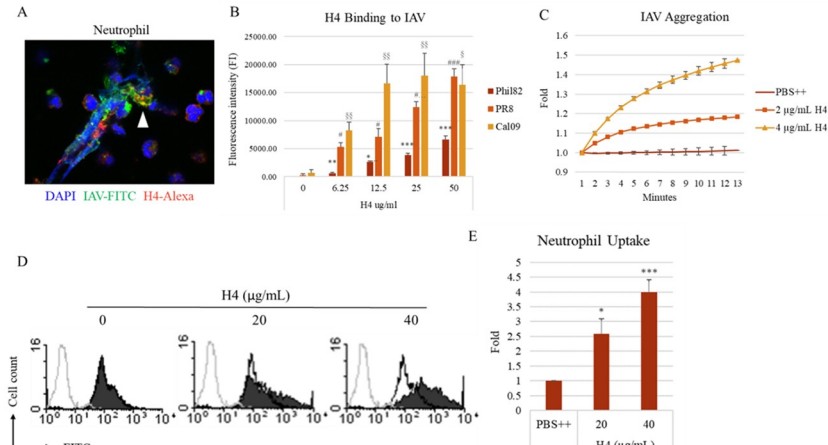

**Fig 1. Histone H4 binds to IAV, causes viral aggregation and increases neutrophil uptake of IAV.** (A) Representative confocal images of neutrophils treated with Phil82 strain of IAV. DNA was stained blue with DAPI, Phil82 IAV green with FITC, and histone H4 red with Alexa Fluor 594. The image shown was taken at 100x magnification. (B) 96-well plates were coated with H4 or BSA and then incubated with Alexa-labeled Phil82, PR8 or Cal09 IAV strains for 45 minutes. Binding of H4 to IAVs was determined by detecting the fluorescence intensity. (C) Viral aggregation was measured by increased light transmission through stirred suspensions of Phil82 IAV. (D) Histone H4 was pre-incubated with FITC-labeled IAV for 30 minutes, and then it was added to the neutrophils for 45 minutes. Uptake of virus by neutrophils was measured by flow cytometry using Trypan blue to quench extracellular fluorescence. Control cells that were not treated with Phil-FITC are shown in each histogram overlay (gray lines). Cells treated only with Phil-FITC but not histone H4 are shown in each histogram overlay (black lines) Cells incubated with virus that was treated with H4 are shown as solid black peaks. (E) Fold of mean fluorescence intensity from flow cytometry is shown (PBS indicates virus alone incubated in PBS without H4). N = 5. Results are presented as mean ± S. E.M (*: $P \leq 0.05$, **: $P \leq 0.01$, ***: $P \leq 0.001$).

representative of three similar experiments. We previously reported that arginine rich histones, especially histone H4, have anti-viral activity against IAV. We now show that histone H4 binds to several IAV strains. Phil82 (seasonal H3N2), PR8 (pandemic H1N1, mouse adapted) or Cal09 (pandemic H1N1) IAV strains were labeled with Alexa and were allowed to bind to plates coated with histone H4 as described in methods (Fig 1B). For subsequent experiments we used the Phil82 IAV strain since we have extensively characterized this strain and its interactions with neutrophils in prior studies. We previously reported that histones cause aggregation of IAV particles with histone H4 being the most potent at this effect at a dose of 10μg/ml. We show in Fig 1C substantial viral aggregation of Phil82 at lower doses of histone H4. Other antimicrobial peptides, like the defensins, not only directly kill bacteria or viruses, but also act as opsonins, promoting uptake of these pathogens by phagocytes. Similarly, we now show that 40 μg/mL histone H4 caused a 4-fold increase in neutrophil uptake of Phil82 IAV compared to PBS control groups (Fig 1D and 1E).

**B. Histone H4 enhances neutrophil respiratory burst responses to IAV.** As shown in Fig 2A, Phil82 IAV (MOI = 40) or histone H4 alone caused significant $H_2O_2$ production (as previously reported—[20, 34]), and pre-incubation of the virus with histone H4 markedly enhanced the response induced by IAV. To determine the role of extracellular and intracellular calcium sources in these responses, we restricted the availability of extracellular calcium by using PBS buffer without calcium and restricted the availability of intracellular calcium by pre-incubating neutrophils with BAPTA-AM to chelate intracellular calcium. We observed less histone H4-induced $H_2O_2$ production without extracellular calcium (Fig 2B). Chelation of intracellular calcium with BAPTA- AM reduced responses to histone H4 or IAV alone

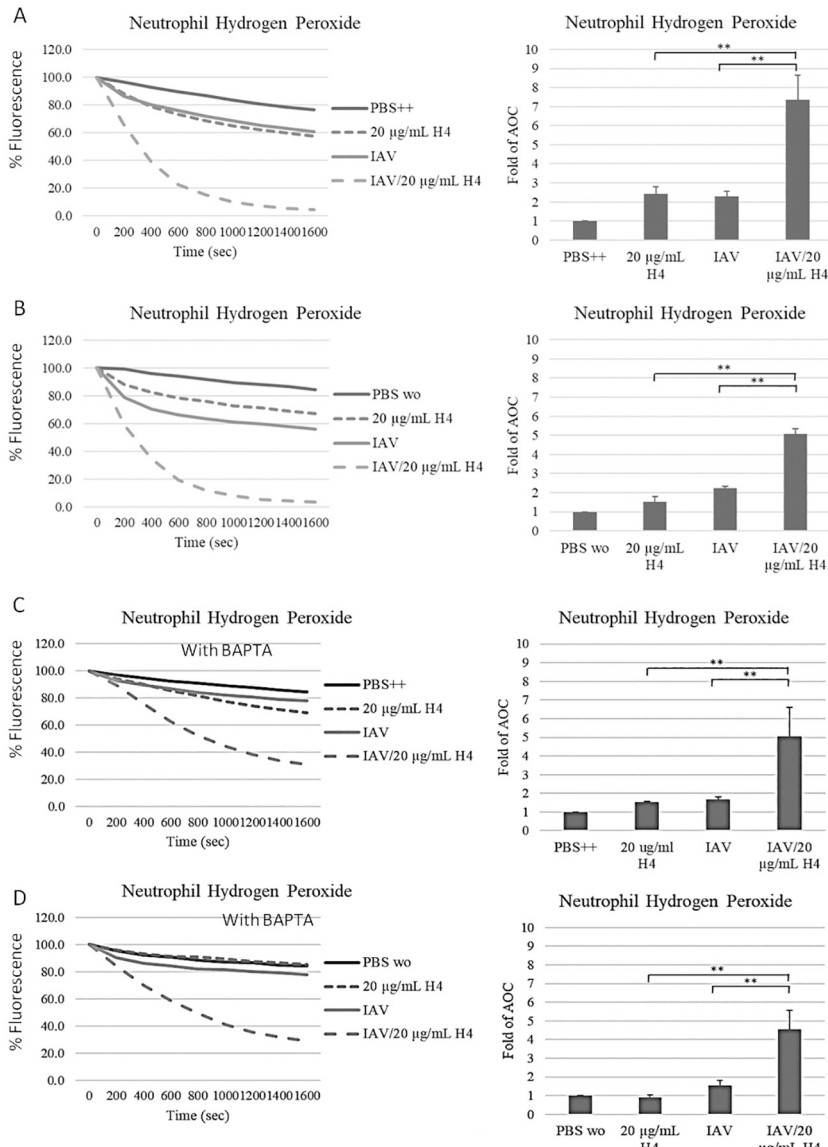

**Fig 2. Histone H4 enhances IAV-induced neutrophil H₂O₂ responses: Calcium dependence.** Human neutrophils were treated with control buffer alone (PBS), H4, IAV or the combination of IAV and H4. Samples were made up either with PBS containing calcium (PBS++) (A and C) or PBS without calcium (PBS wo) (B and D). In panels C and D neutrophils were pre-incubated with 20 µM BAPTA-AM for 10 minutes before the experiments the chelate intracellular calcium. Neutrophils were treated with PBS, histone H4, Phil82 IAV strain or combinations of histone H4 and Phil82 strain. For the combinations, histone H4 and the virus were pre-incubated for 30 minutes prior to the experiment. Hydrogen peroxide production was measured by assessing the reduction in scopoletin fluorescence. Mean fluorescence curves are shown in the left panels, and fold changes of area over the curve (AOC) for those experiments are shown in the right panel. The MOI for these experiments was 40. N = 5. Results are presented as mean ± S.E.M (*: P ≤ 0.05, **: P ≤ 0.01, ***: P ≤ 0.001).

(Fig 2C), and there was no significant H₂O₂ production for either of these stimuli alone when both intra- and extra-cellular calcium were removed (Fig 2D). It was striking, however, that the response to the combination of histone H4 and IAV appeared to be significantly less calcium dependent that either stimulus alone. There was slight reduction in IAV/histone H4 combination-induced H₂O₂ production when we restricted extracellular or intracellular

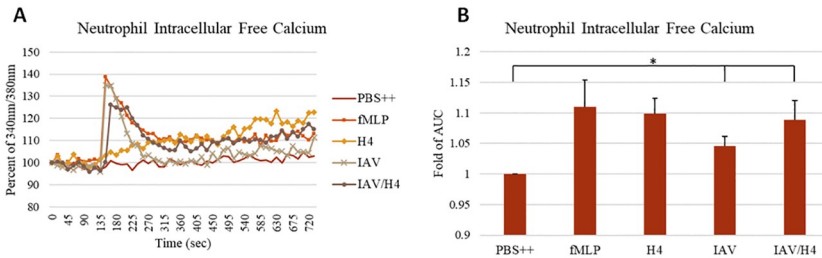

**Fig 3. Intracellular free calcium responses to IAV, H4 or the combination of IAV and H4.** Intracellular free calcium changes in neutrophils treated with PBS, fMLP, histone H4, Phil82 IAV strain or combinations of histone H4 and Phil82 strain were measured by detecting the fluorescence of Fura-2 AM loaded neutrophils. For the combinations, histone H4 and the virus were pre-incubated for 30 minutes prior to the experiment. All samples were added to the cells at 135 seconds. Mean fluorescence curves are shown in panel A and fold changes of area under the curve (AUC) for these experiments are shown panel B. The MOI for these experiments was 40. N = 5. Results are presented as mean ± S.E.M (*: P ≤ 0.05).

calcium sources (Fig 2B and 2C). Furthermore, the combination of IAV and H4 still induced H2O2 production even after removal of both extracellular calcium and chelation of intracellular calcium with BAPTA (Fig 2D).

As shown in Fig 3 and previously reported [20, 32] both histone H4 and IAV caused calcium elevation in neutrophils. The response to IAV alone somewhat resembles the response to fMLP which was tested in parallel. Both stimuli cause and early rise in calcium followed by a recovery close to baseline starting level. However, histone H4 did not cause an initial peak but rather caused a slow progressive rise in intracellular calcium. This rise appears to be due to membrane permeabilization caused by histone H4 [20] and this effect of H4 has been reported for other cell types as well [21, 26]. The combination of H4 and IAV combined features of both stimuli with an early peak but also a persistent elevation at later time points.

## II. CRP and SP-D bind to histone H4 and reduce neutrophil activation by histone H4

**A. CRP and SP-D bind to histone H4.**   As noted, CRP has been reported to counteract effects of H4 on endothelial cells and protect against H4 related mortality in mice. SP-D reduces lung inflammation caused by a variety of insults (including IAV) [38]. We now show that CRP and SP-D both bind to histone H4 by two methods. First we tested binding of CRP, SP-D or SP-D's isolated trimeric neck and carbohydrate binding domain (NCRD) to H4 by ELISA. We used two NCRD preparations including the wild type version and the D325A +R343V double mutant NCRD (*NCRD) which we developed for its increased binding affinity to mannan and IAV [39]. All of these preparations showed significant binding to H4 (Fig 4A–4D). Note that for these experiments background binding to uncoated ELISA plates was subtracted from the results shown.

We also tested binding in solution by incubating the proteins together and then testing if they form a complex large enough to precipitate out of solution after low speed centrifugation (1200 x g). We have previously shown that SP-D can precipitate IAV particles in this manner. After incubation of histone H4 with either CRP, SP-D or wild type NCRD of SP-D significantly greater amounts of the two proteins precipitated out of solution by this assay as illustrated on SDS gels (Fig 5A and 5B). We performed densitometry readings on these gels to obtain qualitative data confirming statistical significance of the results (Fig 5C and 5D). Histone H4 is not glycosylated. We have previously shown, however, that SP-D and its NCRD

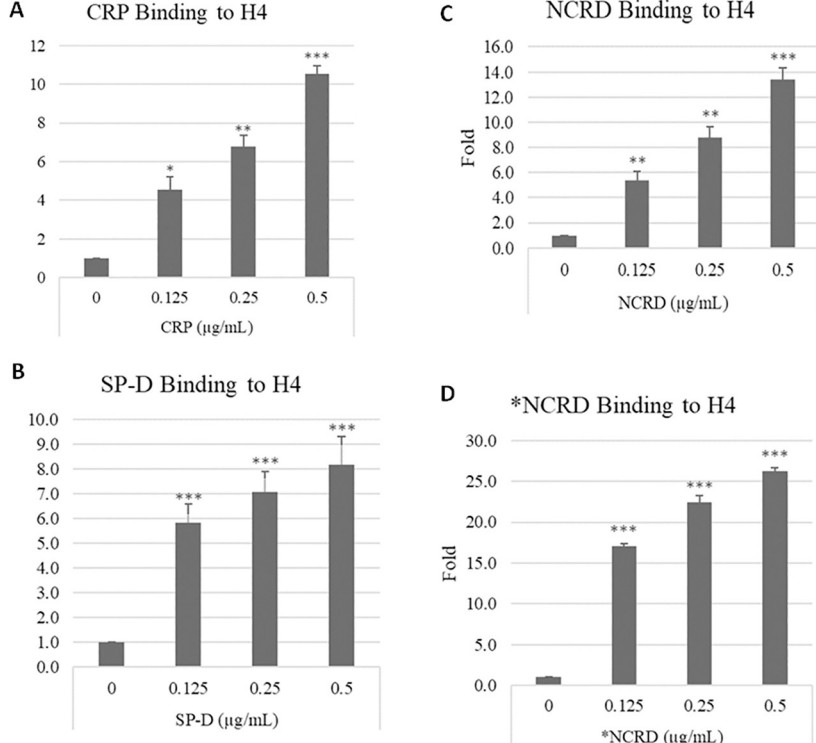

**Fig 4. CRP and SP-D proteins bind to histone H4.** 96-well plates were coated with 10 μg/mL histone H4 overnight. In panel A different concentrations of CRP were added to the plate and the bound CRP was detected by anti-CRP antibody (HRP conjugated) and TMB substrate. Fold changes in light absorption are shown. (B) Different concentrations of SP-D were added to the H4 coated plates and the bound SP-D was detected by anti-SP-D antibody (HRP conjugated) and TMB substrate. Fold changes in light absorption are shown. (C) Different concentrations of NCRD were added to the plate and the bound NCRD was detected by anti-S antibody (HRP conjugated) and TMB substrate. Fold changes in light absorption are shown. (D) Different concentrations of D325A+R343V double mutant NCRD (*NCRD) were added to the H4 coated plates and the bond *NCRD was detected by anti-S antibody (HRP conjugated) and TMB substrate. N = 5. Results are presented as mean ± S.E.M (*: P ≤ 0.05, **: P ≤ 0.01, ***: P ≤ 0.001).

bind to defensins (which are also cationic peptides) based on charge interactions [40–42]. We confirmed that precipitation of H4 by NCRD was not inhibited by EDTA or maltose (which inhibit SP-D or NCRD's calcium-dependent lectin binding to carbohydrates and IAV [33]) (Fig 5D).

**B. Effects of CRP or SP-D proteins on interactions of histone H4 with IAV.**   Pre-incubation of histone H4 with CRP reduced or abolished its effects on IAV (Fig 6), including blocking viral aggregation and viral neutralization by H4 and also reducing the ability of histone H4 to promote uptake of the virus by neutrophils. Of note, CRP did not have any effect on its own in these assays indicating a lack of direct effects on IAV or neutrophils. Since SP-D and the D325A+R343V double mutant NCRD (*NCRD) have strong effects on their own in these assays [22], it was harder to interpret potential effects on histone H4's activities. We used lower doses of histone H4 and SP-D to determine if there were cooperative effects on neutralization or viral aggregation assays. As shown in Table 1, addition of histone H4 did not significantly increase the antiviral activity of SP-D vs. two tested viral strains. Additive effects were found by adding histone H4 to the double mutant *NCRD (Table 2).

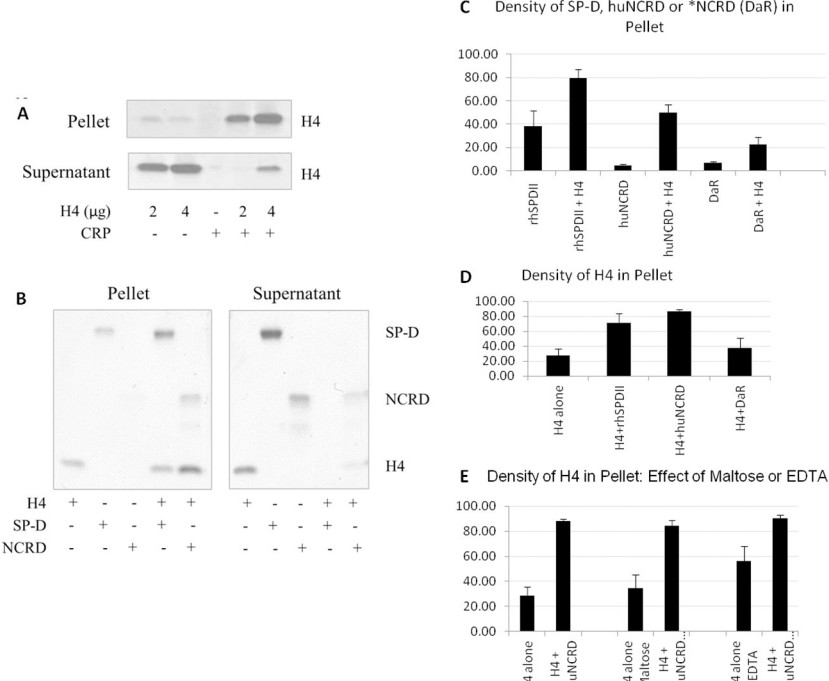

**Fig 5. Binding of CRP, SP-D or NCRD cause precipitation of histone H4 from solution at low speed centrifugation.** (A) 2 μg or 4 μg of histone H4 was incubated with 2 μg of CRP for 30 minutes. After centrifugation, pellet and supernatant from each sample were run on a gel separately and stained with Gelcode blue. A representative result is shown. Similar effects were seen when H4 was pre-incubated with either SP-D or wild type NCRD (B). Densitometry readings were made on pellet bands from gels from 4 replicate experiments and these are shown in panel C for CRP, SP-D and NCRD bands and in panel D for histone bands. The effect of carrying out incubation of NCRD in solution containing EDTA or maltose (10 mM) is shown in panel E. Results of panels C-E are mean ± S.E.M of 4 experiments and in each case significantly more protein (p< 0.05) was found in the pellets when proteins were co-incubated than when single proteins were tested.

**C. CRP and/or SP-D block histone H4-induced neutrophil activation.** As shown in Fig 7, CRP totally blocked hydrogen peroxide production induced by histone H4, while SP-D and the D325A+R343V double mutant NCRD (*NCRD) reduced the response by about 50%. CRP also totally blocked the sustained calcium influx induced by histone H4 (Fig 8A), and SP-D reduced the calcium response by about 50% as well (Fig 8B). However, D325A+R343V double mutant NCRD (*NCRD) did not significantly reduce the histone H4 induced calcium response (Fig 8C). In addition to triggering neutrophil $H_2O_2$ generation we reported that histone H4 causes rapid neutrophil membrane depolarization as well as MPO release and caspase 3 activation over a longer time course [20].

CRP markedly inhibited neutrophil membrane depolarization caused by H4 (Fig 9). Neither CRP nor SP-D induced neutrophil MPO release on their own, but they markedly reduced histone H4-induced release (Fig 10A). D325A+R343V double mutant NCRD (*NCRD) alone increased MPO release, but it reduced by about 50% of neutrophil degranulation induced by histone H4 (Fig 10A). CRP, SP-D and D325A+R343V double mutant NCRD (*NCRD) did not have any effects on neutrophil caspase 3 activation on their own, but CRP totally blocked the caspase 3 activity induced by histone H4. In contrast, SP-D and D325A+R343V double mutant NCRD (*NCRD) did not alter the caspase 3 activation caused by histone H4 (Fig 10B).

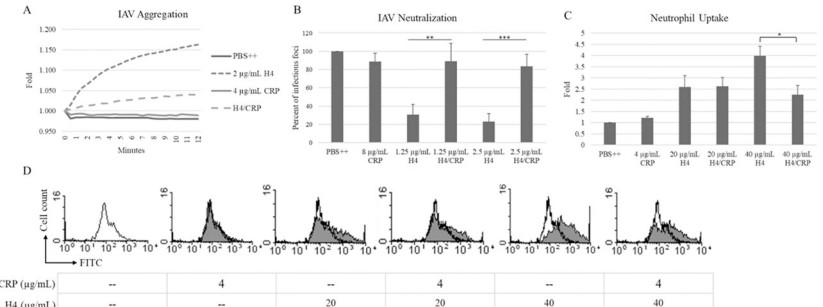

**Fig 6. CRP decreases histone H4-enhanced IAV aggregation, as well as IAV neutralizing activity and increased neutrophil uptake of IAV caused by H4. (A) Viral aggregation**: 2 μg/mL of histone H4 or 4 μg/mL of CRP were incubated alone or together for 30 minutes before the experiments. Viral aggregation was measured by increased light absorption through stirred suspensions of Phil82 IAV. **(B) Viral neutralization**: 1.25 or 2.5 μg/mL of histone H4 or 8 μg/mL of CRP were incubated alone or together for 30 minutes before the experiments, as indicated. IAV neutralization assessed using the fluorescent focus assay for detection of viral nucleoprotein. Phil82 strain was used. **(C, D) Neutrophil uptake of IAV**: 20 or 40 μg/mL of histone H4 or 4 μg/mL of CRP were incubated alone or together for 30 minutes before the experiments. Each sample was pre-incubated with FITC-labeled Phi82 IAV for 30 minutes, and then this was added to the neutrophils for 45 minutes. Uptake of virus by neutrophils was measured by flow cytometry. Cells treated only with Phil-FITC are shown in each histogram overlay (black peaks). Fold of mean fluorescence intensity from flow cytometry is shown in B. N = 5. Results are presented as mean ± S.E.M (*: P ≤ 0.05, **: P ≤ 0.01, ***: P ≤ 0.001).

## Discussion

As noted in the introduction cell free histones behave in many ways like other cationic antimicrobial peptides and have broad spectrum antiviral, antifungal and antibacterial activities [7, 43]. However, there is also abundant evidence that extracellular histones have pro-inflammatory effects that can lead to serious manifestations of sepsis, lung injury and hypercoagulation [43]. In the current paper we extend upon prior observations both in terms of antiviral activity and pro-inflammatory effects of histone H4. We focused on histone H4 since it had the most potent antiviral activity among histones, it has been implicated specifically in pro-inflammatory effects in vivo, and also for the practical reason that the commercially available preparation of histone H4 is free of endotoxin, whereas other purified histones we tested were not. Abrams et al. investigated 250 patients with severe trauma and found that circulating histone

**Table 1. Effects of SP-D or H4 alone or in combination on infectivity of IAV.**

| | H4 Conc. ng/ml | 0 | 75 | 156 | 312 | 624 |
|---|---|---|---|---|---|---|
| **SP-D Conc.ng/ml** | Viral Strain | | | | | |
| **0** | Phil 82 | 100 | 75±8 | 69±2 | 57±4 | 61±6 |
| **0** | Aichi 68 | 100 | 78±5 | 59±7 | 53±10 | 45±6 |
| **10** | Phil 82 | 67±7 | 58±9 | 60±6 | | |
| **10** | Aichi 68 | 76±7 | 60±5 | 59±6 | | |
| **20** | Phil 82 | 51±8 | 44±10 | 38±12 | | |
| **20** | Aichi 68 | 61±9 | 57±9 | 54±9 | | |
| **40** | Phil 82 | 45±7 | 43±9 | 37±10 | | |
| **40** | Aichi 68 | 49±8 | 47±11 | 48±9 | | |

N = 5, P<0.05 for all compared to control but addition of H4 did not significant reduce infectious foci compared to SP-D alone.

**Table 2. Effects of huNCRD with D325A+R343V substitutions (\*NCRD) or H4 alone or in combination on infectivity of IAV.**

| | H4 Conc. ng/ml | 0 | 75 | 156 | 312 | 624 |
|---|---|---|---|---|---|---|
| **\*NCRD Conc.ng/ml** | Viral Strain | | | | | |
| **0** | Phil 82 | 100 | 68±8 | 49±6 | 39±3 | 37±4 |
| **0** | Aichi 68 | 100 | 72±2 | 65±3 | 50±5 | 47±5 |
| **125** | Phil 82 | 60±7 | 39±7* | 34±4* | | |
| **125** | Aichi 68 | 86±3 | 57±4* | 47±4* | | |
| **250** | Phil 82 | 52±9 | 37±5 | 27±4* | | |
| **250** | Aichi 68 | 66±5 | 54±6 | 39±3* | | |
| **500** | Phil 82 | 40±4 | 27±5* | 24±3* | | |
| **500** | Aichi 68 | 50±3 | 40±3* | 33±2* | | |

N = 5 for Phil 82 and n-6 for Aichi 68; p<0.05 for all compared to control;

\* significantly reduced compared to H4 or \*NCRD alone by ANOVA.

levels ranged from 10 to 230 μg/ml within 4 h after injury, and the level peaked at 24 h and remained detectable after 72 h [9]. In our study, we used histone H4 up to 40 μg/ml, which is within this range. We found that histone H3 (the other arginine rich histone) had similar effects as H4, but we did not pursue these findings further due to presence of endotoxin in available histone H3 preparations. Histone H4 has been specifically shown to cause membrane permeabilization in endothelial cells and smooth muscle cells [21, 26]. We made similar findings with histone H4 and neutrophils and showed that this membrane permeabilization leads to membrane depolarization and calcium influx which in turn triggers a respiratory burst response, adhesion, IL-8 and MPO release [20].

Histone H4 binds to IAV and causes viral neutralization and viral aggregation ([7] and this paper). It is likely that IAV encounters histones during infection. Using confocal microscopy we show binding of histone H4 in neutrophil extracellular traps induced by and containing IAV. We show that histone H4 strongly modulates interactions of IAV with neutrophils most notably increasing viral uptake by these cells and markedly potentiating the neutrophil respiratory burst response to the virus. While some of these activities might be construed to be protective, evidence from in vivo studies suggest that the predominant effect of extracellular histones during IAV infection is potentiation of harmful inflammation. Ashar et al. found that

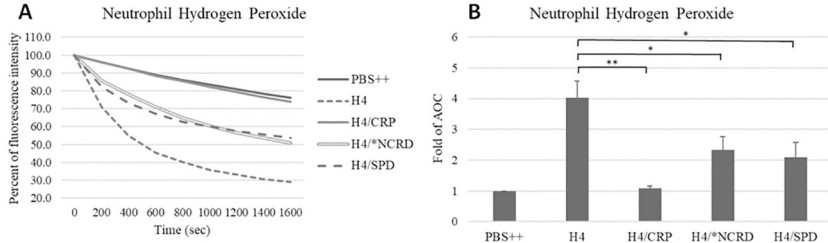

**Fig 7. Effects of CRP and SP-D on histone H4-induced neutrophil hydrogen peroxide responses.** Hydrogen peroxide production in neutrophils on exposure to PBS++ control buffer, fMLP, or 40 μg/mL histone H4 preincubated with or without 80 μg/mL CRP, 8 μg/mL D325A+R343V double mutant NCRD (\*NCRD) or 16 μg/mL SP-D was measured by assessing the reduction in scopoletin fluorescence. Panel A shows mean scopoletin fluorescence and Panel B show fold changes of area over the curve (AOC).

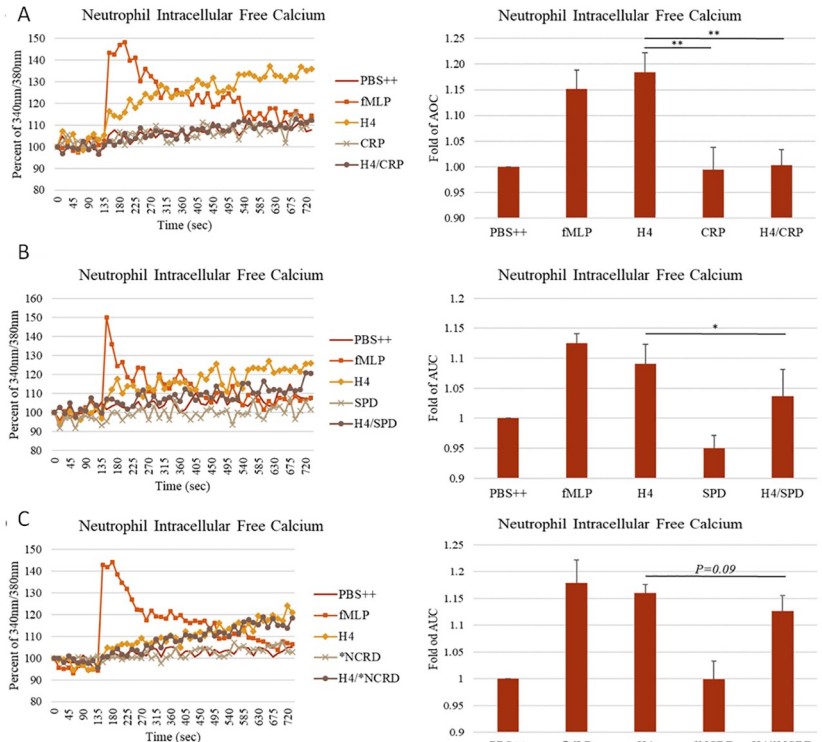

**Fig 8. Effects of CRP or SP-D on neutrophil intracellular calcium responses to H4.** Mean fluorescence curves are shown in the left panels, and fold changes of area under the curve (AUC) for these experiments are shown on the right. Intracellular free calcium was detected by loading neutrophils with Fura-2 AM. (A) Intracellular free calcium changes in neutrophils on exposure to PBS++ control buffer, fMLP, 40 μg/mL histone H4, 80 μg/mL CRP or the combination of histone H4 and CRP. (B) Intracellular free calcium changes in neutrophils on exposure to PBS++ control buffer, fMLP, 40 μg/mL histone H4, 16 μg/mL SP-D or the combination of histone H4 and SP-D. (C) Intracellular free calcium changes in neutrophils on exposure to PBS++ control buffer, fMLP, 40 μg/mL histone H4, 8 μg/mL D325A +R343V double mutant NCRD (*NCRD) or the combination of histone H4 and mutant.

treatment with histones exacerbated lung pathology due to their cytotoxic and coagulatory effects which caused severe lung damage and microvascular thrombosis in mice infected with sub-lethal amounts of pandemic IAV [6]. They also showed that IAV induces NET production and release of histones in lungs of infected mice, and that anti-histone antibodies were protective. We propose that the marked potentiation of neutrophil respiratory burst response to IAV caused by histone H4 would have significant pro-inflammatory consequences in vivo. There is evidence from various murine studies that oxidant production is harmful during IAV infection [44–46].

Several inhibitors of extracellular histones have been reported, such as activated protein C (aPC), heparin, recombinant thrombomodulin (rTM), CRP and pentraxin 3 (PTX3) [43]. Each of these inhibitors have been shown to block aspects of histone-mediated damage. Most of the studies were evaluating their ability of blocking thrombosis and endothelial cytotoxicity. To treat histone-exaggerated lung damage during IAV infection, we were looking for inhibitors blocking interactions between histones and neutrophils. CRP is an acute phase response protein that is upregulated in inflammatory conditions or infections, including IAV infection. CRP levels in IAV-infected patients have been found to range from 34 to 111 mg/L [47]. Serum CRP levels have been correlated with the severity of IAV infection symptoms or fatality

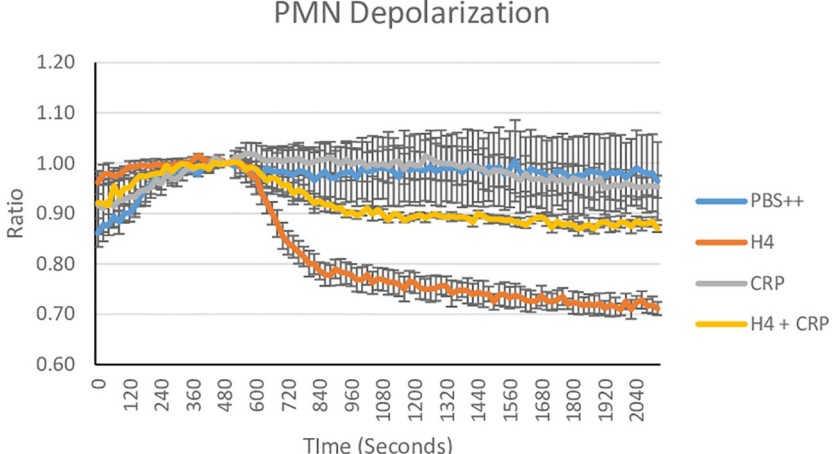

**Fig 9. Effects of SP-D and CRP on neutrophil membrane depolarization caused by histone H4.** Neutrophils were pre-incubated with 50 mM Di-OC5 5 minutes followed by addition of 40 μg/ml of H4 at time zero. Histone H4 alone or histone H4 which had been pre-incubated with CRP was added to neutrophils and depolarization of the neutrophil membrane was indicated by reduction in fluorescence. Results are mean ± SEM for 4 experiments with p< 0.05 when comparing H4 alone to H4 plus CRP (ANOVA).

of IAV-infected patients [48]. CRP is also a marker of severe COVID-19 illness. CRP has been shown to reduce histone-mediated cytotoxicity toward endothelial cells through preventing histones from integrating into cell membranes [21]. Here we show that CRP bound to histone H4 and significantly blocked histone H4-induced neutrophil $H_2O_2$ production, calcium influx and degranulation, suggesting that it may downregulate the excessive neutrophil responses induced by histone H4. By blocking binding of histone H4 to neutrophils CRP also prevents neutrophil membrane permeabization based on the membrane depolarization assay. We have previously presented data that this permeabilization is the key step in histone H4 induced neutrophil activation [20]. On the other hand, CRP also reduced the IAV neutralizing and aggregating activity of histone H4 and the ability of histone H4 to increase neutrophil uptake of IAV. Future *in vivo* studies are needed to confirm whether CRP ameliorates histone-mediated lung damage in response to IAV infection or other inflammatory stimuli.

SP-D is a member of the collectin family responsible for host defense in the lung, and it contributes to the clearance and neutralization of respiratory viruses [22]. SP-D has potent anti-viral and aggregating effects on seasonal IAV, but pandemic strains are resistant to inhibition by SP-D. We previously reported that combinatorial mutations at the 325 and 343 positions (D325A+R343V) in neck and carbohydrate recognition domain (NCRD) of human SP-D improved the binding ability to IAV and thus the mutant NCRD (*NCRD) had greater anti-viral activity against both seasonal and pandemic IAV *in vitro* and *in vivo* [29, 30]. Here we show that SP-D, NCRD and double mutant NCRD (*NCRD) bound to histone H4 and act as histone H4 inhibitors. Binding was not mediated by the calcium dependent lectin activity of SP-D or the NCRDs. Further experiments could be done to clarify the mechanisms of binding of CRP or SP-D to H4. In prior experiments we found that SP-D and NCRD bind to defensins through charge interactions [42], so this may be the case for H4 as well since defensins and H4 are both cationic proteins. Prior studies have shown that SP-D binds to neutrophil NETs [25], which we speculate could in part be related to binding to NET associated histones. *In vivo* studies are needed to confirm whether treatment with SP-D or double mutant NCRD

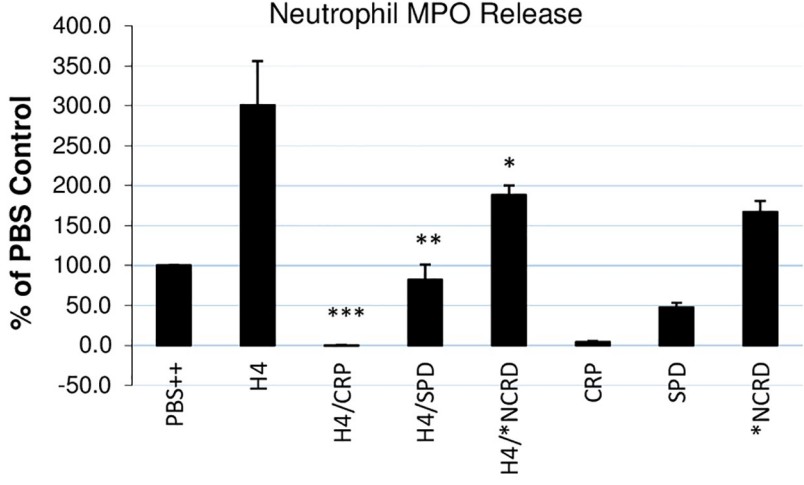

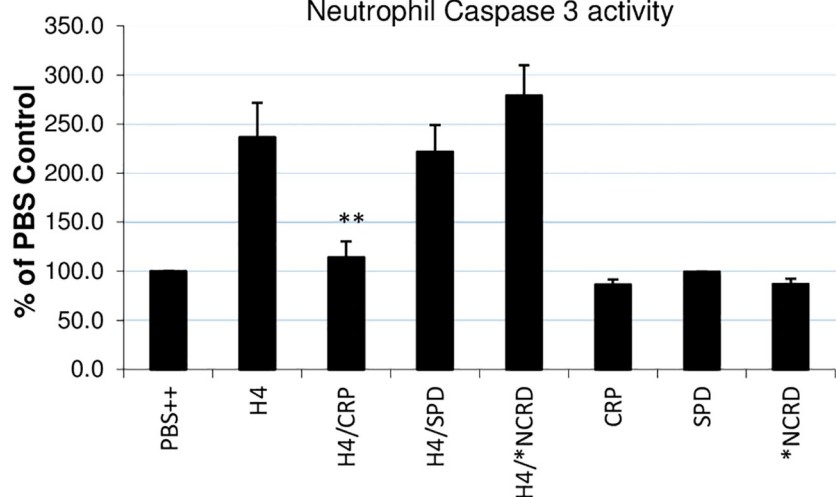

**Fig 10. Effects of CRP or SP-D on neutrophil MPO release or caspase 3 activation by H4.** Neutrophils were treated with PBS++ control buffer, 40 μg/mL histone H4 alone or histone that had been pre-incubated with either CRP, SP-D, or 8 μg/mL D325A+R343V double mutant NCRD (*NCRD). In addition, CRP, SP-D or *NCRD were added alone. Panel A shows effects of the treatments on MPO release into the cell supernatant and panel B shows effects of the treatments on caspase 3 activity as measured with the Z-DEVD–AMC substrate. N = 5. Results are presented as mean ± S.E.M (*: P ≤ 0.05, **: P ≤ 0.01, ***: P ≤ 0.001).

(*NCRD) attenuates histone-mediated lung damage in response to IAV infection. Mice lacking SP-D or SP-A have increased inflammatory responses to IAV infection and other stimuli [38, 49–51]. It is possible that this is in part mediated by the loss of SP-D's ability to reduce histone or neutrophil NET mediated injury. An excellent recent paper showed that SP-D inhibits NET formation in mice in response to LPS and that SP-D knockout mice have increased NET induction by LPS compared to control mice [52]. SP-D and SP-A have been shown to protect against oxidant injury in other contexts [53–55]. Further studies examining binding of SP-A to histones would be of interest as well.

The role of neutrophils in IAV infection is complex. Neutrophils are the first cells recruited in abundance to the IAV infected airway and appear to play an important role in initial viral containment and facilitating the next levels of immune response. Neutrophils can promote

adaptive response development and also limit viral replication and more severe inflammation in some studies [56–59]. Neutrophils can take up IAV and aid in clearance of the virus [41]. In contrast, excessive neutrophilic influx during severe IAV induced pneumonia (e.g. as seen with the highly pathogenic 1918 or avian strains like H5N1) appears to be harmful. A paper by Brandes et al provides a potential explanation for conflicting data on the role of neutrophils in severe IAV infection [60]. They performed a systems analysis of milder or more severe IAV infection in mice and showed that a feed forward circuit can develop in severe, lethal infection which is driven by neutrophils and related cytokines leading to inflammatory injury and death. Whereas full ablation of all neutrophils was found to compromise outcomes in studies by Tate et al [58], Brandes et al performed more limited inhibition of neutrophil recruitment through low dose anti-neutrophil mAb treatment or reduction of HIF signaling and showed improved outcomes without increase in viral loads. HIF signaling was noted as a key driver of myeloid driven inflammation. These results suggest that a partial or modulated downregulation of neutrophil or myeloid activity may be an effective approach. Of interest they found that early it was failure of early control of viral replication that led to development of the excessive neutrophilic inflammation. There is evidence too that severe neutrophil influx and NET formation may be adverse in severe COVID19 illness [3, 4].

Hence, in both severe IAV and COVID-19 histones may play a significant role. There is an additional particular feature of SARS-CoV2 involving likely infection and perturbation of endothelial cells [61, 62]. This feature was found to distinguish lung pathology of fatal COVID-19 from fatal IAV pneumonia [62]. Another major feature of histone toxicity is triggering of the coagulation cascade [43]. Both SARS-CoV1 (and likely SARS-CoV2) and IAV trigger NLRP3 inflammasome activation which is associated with pyroptotic cell death and IL-1 generation [63]. This process could also possibly result in release of intracellular histones. We found that histone H4 causes release of IL-1 and TNF from human monocyte/macrophages which could result in further neutrophil recruitment in vivo [20]. The effects of CRP and SP-D on monocyte/macrophage activation by histones is a good topic for future study. Overall it appears that CRP and SP-D may be part of the innate defense against toxicity of cell free histones in severe lung viral infections. Hence, measures to block action of histones through medical interventions (e.g. antibodies or other means like engineered collectins) could be beneficial.

## Supporting information

**S1 File.**
(PDF)

## Author Contributions

**Conceptualization:** Mitchell White, Kevan Hartshorn.

**Data curation:** I-Ni Hsieh, Marloes Hoeksema, Xavier Deluna.

**Formal analysis:** I-Ni Hsieh, Mitchell White, Kevan Hartshorn.

**Funding acquisition:** Kevan Hartshorn.

**Investigation:** I-Ni Hsieh, Mitchell White, Marloes Hoeksema, Xavier Deluna.

**Supervision:** Kevan Hartshorn.

**Writing – original draft:** I-Ni Hsieh.

**Writing – review & editing:** Kevan Hartshorn.

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
