## [Decision Letter · Decision Letter 0]

7 Jan 2021

PONE-D-20-38304

Histone H4 potentiates neutrophil inflammatory responses to influenza A virus: down-modulation by H4 binding to C-reactive protein and Surfactant protein D.

PLOS ONE

Dear Dr. Hartshorn,

Thank you for submitting your manuscript to PLOS ONE. After careful consideration, we feel that it has merit but does not fully meet PLOS ONE’s publication criteria as it currently stands. Therefore, we invite you to submit a revised version of the manuscript that addresses the points raised during the review process.

I recommend addressing the points highlighted by the referees.

We look forward to receiving your revised manuscript.

Kind regards,

Nades Palaniyar, MSc., PhD.

Academic Editor

PLOS ONE

Journal Requirements:

3.PLOS ONE now requires that authors provide the original uncropped and unadjusted images underlying all blot or gel results reported in a submission’s figures or Supporting Information files. This policy and the journal’s other requirements for blot/gel reporting and figure preparation are described in detail at https://journals.plos.org/plosone/s/figures#loc-blot-and-gel-reporting-requirements and https://journals.plos.org/plosone/s/figures#loc-preparing-figures-from-image-files. When you submit your revised manuscript, please ensure that your figures adhere fully to these guidelines and provide the original underlying images for all blot or gel data reported in your submission. See the following link for instructions on providing the original image data: https://journals.plos.org/plosone/s/figures#loc-original-images-for-blots-and-gels.

Reviewers' comments:

Reviewer's Responses to Questions

**Comments to the Author**

1. Is the manuscript technically sound, and do the data support the conclusions?

Reviewer #1: Yes

Reviewer #2: Yes

2. Has the statistical analysis been performed appropriately and rigorously? 

Reviewer #1: Yes

Reviewer #2: Yes

3. Have the authors made all data underlying the findings in their manuscript fully available?

Reviewer #1: Yes

Reviewer #2: No

4. Is the manuscript presented in an intelligible fashion and written in standard English?

Reviewer #1: Yes

Reviewer #2: Yes

5. Review Comments to the Author

Reviewer #1: The manuscript attempts to unravel the impact of H4 histone on the activation of neutrophils in the context of IAV infection. Importantly, authors reveal the molecular interaction between H4 and CRP as well as H4 and SP-D and their inhibitory effect on H4 induced neutrophil activation and oxidative burst.

The authors did not comparatively evaluate CRP or SP-D binding to H4 and whether they compete for the same binding site on H4.

Also, authors have not discussed the impact of these molecular interactions together on the infectious foci of IAV in vitro and neutrophil uptake of IAV.

Discussion needs to mention the physiological concentrations of these ligands and H4 in BAL of IAV infected individuals and how is it likely to affect the IAV replication. Most of the discussion is focused on the ability of SP-D and CRP on the H4 induced neutrophil activation. Both H4 and SP-D have antiviral effects. Once they bind to each other, it was expected to see an increase in the viral foci. Authors may offer an explanation in this regard. Is it likely that H4 bound to SP-D may still show antiviral effects? or SP-D bound to H4 may still show antiviral effects?

In the Fluorescent focus assay of IAV infectivity, the authors only mention MOI. It would be relevant to mention the number of viral units and the number of cells. Importantly, also indicate how the viral units were enumerated.

Ethics statement may mention the study approval date and duration.

Reviewer #2: The authors show the effect of histone H4 on the activation of neutrophils and promoting neutrophils’ responses that lead to a respiratory burst, and how H4 can exacerbate the inflammatory response of neutrophils to influenza A virus (IAV). Interestingly, they demonstrate that SP-D or CRP can inhibit the pro-inflammatory effect of H4 in neutrophils.

The results and data provided by the authors are interesting and novel, besides the demonstrated role of CRP and SP-D to reduce the H4 pro-inflammatory response in neutrophils in-vitro is relevant and the first step to perform future in vivo experiments. However, some additional details of the experiments or information should be provided, as well as explanation or comments in some results.

Specific comments:

1. The methods section should be carefully reviewed. More details should be provided, as well as the method details in the figure legends. Authors should provide buffer protein compositions (e.g. for all the recombinant SP-Ds used and commercial H4 and CRP). What is control buffer? and if it is PBS++, it should be stated it in methods, sometimes PBS++ is also indicated in the figure legend and others isn’t – revise figure legends. Concentrations of reagents should be provided: e.g. how much calcium was added to PBS? Incubations of peptides with the neutrophils were performed at 37 ˚C or at 37 ˚C in an incubator with 5% (v/v) CO2? Same comment applies for conditions when “treatments” are performed e.g. in the measurement of caspase 3 activity it says “human neutrophils were treated with indicated proteins for 5 hours” (where? Which temperature?).

2. Production of recombinant human SP-D in CHO cells yield a combination of different oligomeric forms of SP-D. Therefore, it should be specified if rhSP-D dodecamers were isolated from the mixture of oligomers obtained and purified from the CHO cells. Otherwise, it should be indicated as rhSP-D without specifying the oligomeric form or emphasizing that is the most abundant form but not 100%.

3. In Figure 2, the graphs for A-B seem identical to the graphs for C-D but for the label “with BAPTA”. Authors should double check if they copied the right graphs to figure 2. It seems that C-D might be the wrong ones and A-B were taken again by mistake.

4. The authors showed binding of CRP and SP-D to H4 with two different methods. There is no question that binding takes place between CRP and H4 and also between SP-D and H4. However, the second method applied, the co-precipitation by centrifugation is somehow intriguing. The gels show that the method is working and reporting binding like the ELISAs. It is very surprising that the bound complex of SP-D (especially the NCRD) and H4 (a 12 kDa protein) precipitates with a centrifugation at 1,200 x g for 5 min. Ultracentrifugation at 100,000 x g is performed to pellet pulmonary surfactant from bronchoalveolar lavages and SP-D is recovered in the supernatant instead of the pellet (Taeusch, H. W., J. Bernardino de la Serna, et al. 2005. Biophys. J.), in addition, in SP-D purification from BAL or amniotic fluid a centrifugation step at 2,000 to 10,000 x g depending on the paper is performed (e.g. Leth-Larsen, R., et al., 1999. Biochem J; Strong P., et al., 1998. J of Immunological Methods). Therefore, it is surprising that the complex SP-D-H4 is pelleting at 1,200 x g. Is a visible pellet obtained in that centrifugation? How much are the volumes for each reagent and final volume that allow to differentiate and pipet supernatant and pellet?

I would recommend the authors to perform the binding experiments in presence of EDTA and maltose (Figure 5E, which in the text is pointed as 5D -see last figure mentioned in that paragraph in page 14) by the ELISA method as a more robust technique to confirm that the binding is not calcium dependent.

The authors show that the binding site of SP-D to H4 is not in the collagen or N-terminal domain because binding is observed with the NCRD mutant. At the same time, binding is reported to be non-calcium dependent and non-CRD mediated. However, looking at the ELISA results (Figure 4) and comparing the binding of NCRD and *NCRD (the mutant with increased affinity to mannan) to H4, it is intriguing that the *NCRD mutant seems to bind more or with higher affinity to H4 than the NCRD peptide, taking into account that the binding is not dependent of the lectin activity of the protein. How do the authors explain it? Could it be related to the charge variation also induced by the amino-acids that are substituted in the mutant?

5. A significant inhibitory effect of CRP/H4 in neutrophil uptake of IAV is observed at 40 µg/mL of H4 in combination with PCR, but inhibition is not observed at 20 µg/mL of H4 at the same CRP concentration. Do the authors have a proposed explanation? Why at a lower concentration of H4, there isn’t an inhibitory effect of CPR in neutrophil uptake of IAV? (In addition, review the figure legend of figure 6 C-D, since the concentration of H4 indicated does not match the figure and does not indicate 2 different testing concentrations, besides PCR concentration is 8 µg/mL when in the following experiments is 80 µg/mL).

6. In table 1 and 2, why higher concentrations (ng/mL) were tested for the mutant NCRD* than for SP-D. Is it the effect observed with the mutant NCR* due to the higher concentrations of protein there in comparison to SP-D (full length)?

7. The reason behind the selection of the different concentrations of CRP, NCRD* and SP-D in the experiments in combination with H4 (results section B and C) should be, at a minimum, indicated or commented in the discussion. How the authors explain the different effects observed between SP-D and NCRD* in intracellular free calcium and MPO release?

Statements as “data not shown” should be avoided and the data/figure provided in supplemental material.

8. Are any of the peptides (CRP, H4, SP-D, NCRD and NCRD*) in an EDTA-containing buffer? In case they are, is the EDTA concentration being compensated with additional calcium before being added to the cells to exclude any EDTA-related effect in the experiments (for example in the MPO release, figure 10)?

9. Recently, it has been shown that SP-D reduces LPS-induced NETosis in vivo and the detrimental effect of NETs in pulmonary surfactant (in mice lacking SP-D) (Arroyo, R., et al., 2020, Comm Biology), which should be referred in the discussion of SP-D ability to reduce NET mediated injury if authors want to discuss that idea.

6. PLOS authors have the option to publish the peer review history of their article (what does this mean?). If published, this will include your full peer review and any attached files.

Reviewer #1: **Yes: **Taruna Madan

Reviewer #2: **Yes: **Raquel Arroyo

---

## [Author Response · Author response to Decision Letter 0]

7 Feb 2021

Response to reviews: 

General editors review: Data not shown not acceptable.

 We have removed the reference to data not shown since that data (viral aggregation by SP-D combined with H4) was not vital to the manuscript. 

Reviewer #1: The manuscript attempts to unravel the impact of H4 histone on the activation of neutrophils in the context of IAV infection. Importantly, authors reveal the molecular interaction between H4 and CRP as well as H4 and SP-D and their inhibitory effect on H4 induced neutrophil activation and oxidative burst.

The authors did not comparatively evaluate CRP or SP-D binding to H4 and whether they compete for the same binding site on H4.

Also, authors have not discussed the impact of these molecular interactions together on the infectious foci of IAV in vitro and neutrophil uptake of IAV.

The impact of the interaction of H4 and CRP on the antiviral activity and neutrophil uptake activity of H4 is shown in figure 6. CRP decreased both of these activities of H4. The effect of SP-D on neutralization and neutrophil uptake by H4 was more complex to evaluate since both have the ability to reduce viral infectivity and increase neutrophil uptake of virus. As shown in tables 1 and 2 the combination of SP-D and H4 overall showed about the same neutralizing activity as SP-D alone. Hence H4 did not reduce the potent neutralizing activity of SP-D (although it also did not create additive activity). 

Discussion needs to mention the physiological concentrations of these ligands and H4 in BAL of IAV infected individuals and how is it likely to affect the IAV replication. Most of the discussion is focused on the ability of SP-D and CRP on the H4 induced neutrophil activation. Both H4 and SP-D have antiviral effects. Once they bind to each other, it was expected to see an increase in the viral foci. Authors may offer an explanation in this regard. Is it likely that H4 bound to SP-D may still show antiviral effects? or SP-D bound to H4 may still show antiviral effects?

Estimated levels of H4 found in BAL in other inflammatory states has been added to the discussion and is in the range of amounts tested in our paper. We cannot easily explain how SP-D can bind to H4 and still retain antiviral activity as shown in Tables 2 and 3. We have added to the discussion on this. 

In the Fluorescent focus assay of IAV infectivity, the authors only mention MOI. It would be relevant to mention the number of viral units and the number of cells. Importantly, also indicate how the viral units were enumerated.

The viral units were enumerated by infectious focus assay which therefore indicates units of infectious virus. We cannot exclude that some non-infectious particles might have been present. There are 5000 cells in each well of the 96 well plate and various dilutions of virus are added and infected cells counted before the virus has a chance to spread to adjacent cells. We then can correct for the dilution factor of virus. Hence if we count a total of 50 particles in the well and the dilution was 1:100 then this amount of virus would give an MOI of one. 

Ethics statement may mention the study approval date and duration.

The IRB approval is renewed every year in June and involves permission to obtain blood for isolation of neutrophils, serum and monocytes from peripheral blood of healthy donors. 

Reviewer #2: The authors show the effect of histone H4 on the activation of neutrophils and promoting neutrophils’ responses that lead to a respiratory burst, and how H4 can exacerbate the inflammatory response of neutrophils to influenza A virus (IAV). Interestingly, they demonstrate that SP-D or CRP can inhibit the pro-inflammatory effect of H4 in neutrophils.

The results and data provided by the authors are interesting and novel, besides the demonstrated role of CRP and SP-D to reduce the H4 pro-inflammatory response in neutrophils in-vitro is relevant and the first step to perform future in vivo experiments. However, some additional details of the experiments or information should be provided, as well as explanation or comments in some results.

Specific comments:

1. The methods section should be carefully reviewed. More details should be provided, as well as the method details in the figure legends. Authors should provide buffer protein compositions (e.g. for all the recombinant SP-Ds used and commercial H4 and CRP). What is control buffer? and if it is PBS++, it should be stated it in methods, sometimes PBS++ is also indicated in the figure legend and others isn’t – revise figure legends. Concentrations of reagents should be provided: e.g. how much calcium was added to PBS? Incubations of peptides with the neutrophils were performed at 37 ˚C or at 37 ˚C in an incubator with 5% (v/v) CO2? Same comment applies for conditions when “treatments” are performed e.g. in the measurement of caspase 3 activity it says “human neutrophils were treated with indicated proteins for 5 hours” (where? Which temperature?).

These details have been added to the methods. All experiments were performed in PBS with calcium and magnesium (indicated as PBS++) except where it is noted that PBS without calcium and magnesium were used. All incubations with cells were done in a humidified incubator at 37C and with 5% CO2. 

2. Production of recombinant human SP-D in CHO cells yield a combination of different oligomeric forms of SP-D. Therefore, it should be specified if rhSP-D dodecamers were isolated from the mixture of oligomers obtained and purified from the CHO cells. Otherwise, it should be indicated as rhSP-D without specifying the oligomeric form or emphasizing that is the most abundant form but not 100%.

We have highlighted in methods that dodecameric SP-D was used in these studies and that the NCRD proteins were trimers. 

3. In Figure 2, the graphs for A-B seem identical to the graphs for C-D but for the label “with BAPTA”. Authors should double check if they copied the right graphs to figure 2. It seems that C-D might be the wrong ones and A-B were taken again by mistake.

We thank the reviewer very much for noticing this mistake and have replaced parts C and D of figure 2 with the correct panels. 

4. The authors showed binding of CRP and SP-D to H4 with two different methods. There is no question that binding takes place between CRP and H4 and also between SP-D and H4. However, the second method applied, the co-precipitation by centrifugation is somehow intriguing. The gels show that the method is working and reporting binding like the ELISAs. It is very surprising that the bound complex of SP-D (especially the NCRD) and H4 (a 12 kDa protein) precipitates with a centrifugation at 1,200 x g for 5 min. Ultracentrifugation at 100,000 x g is performed to pellet pulmonary surfactant from bronchoalveolar lavages and SP-D is recovered in the supernatant instead of the pellet (Taeusch, H. W., J. Bernardino de la Serna, et al. 2005. Biophys. J.), in addition, in SP-D purification from BAL or amniotic fluid a centrifugation step at 2,000 to 10,000 x g depending on the paper is performed (e.g. Leth-Larsen, R., et al., 1999. Biochem J; Strong P., et al., 1998. J of Immunological Methods). Therefore, it is surprising that the complex SP-D-H4 is pelleting at 1,200 x g. Is a visible pellet obtained in that centrifugation? How much are the volumes for each reagent and final volume that allow to differentiate and pipet supernatant and pellet?

I would recommend the authors to perform the binding experiments in presence of EDTA and maltose (Figure 5E, which in the text is pointed as 5D -see last figure mentioned in that paragraph in page 14) by the ELISA method as a more robust technique to confirm that the binding is not calcium dependent.

The authors show that the binding site of SP-D to H4 is not in the collagen or N-terminal domain because binding is observed with the NCRD mutant. At the same time, binding is reported to be non-calcium dependent and non-CRD mediated. However, looking at the ELISA results (Figure 4) and comparing the binding of NCRD and *NCRD (the mutant with increased affinity to mannan) to H4, it is intriguing that the *NCRD mutant seems to bind more or with higher affinity to H4 than the NCRD peptide, taking into account that the binding is not dependent of the lectin activity of the protein. How do the authors explain it? Could it be related to the charge variation also induced by the amino-acids that are substituted in the mutant?

We were also surprised that the CRP and SP-D caused precipitation of the H4 at low speed centrifugation. We have found that SP-D causes precipitation of virus as well in similar studies in the past. Since both SP-D and CRP are multimeric proteins we speculate that they are able to cross-link other proteins to form large complexes. This could facilitate clearance of harmful proteins (like free histones) through muco-ciliary clearance perhaps. The effect of the NCRD trimers is more difficult to explain. However, we have shown that such trimers can cross-link and aggregate viral particles depending on the orientation of the three binding sites on the trimer with respect to viral hemagglutinin (see Goh et al Biochemistry. 2013 Nov 26;52(47):8527-38). 

It is possible as noted by the reviewer that binding of SP-D to H4 is based on charge interactions since we have shown a similar mechanism of binding or SP-D or NCRD to defensins (also cationic proteins) in prior studies (ref 42). This has been noted in the discussion.

5. A significant inhibitory effect of CRP/H4 in neutrophil uptake of IAV is observed at 40 µg/mL of H4 in combination with PCR, but inhibition is not observed at 20 µg/mL of H4 at the same CRP concentration. Do the authors have a proposed explanation? Why at a lower concentration of H4, there isn’t an inhibitory effect of CPR in neutrophil uptake of IAV? (In addition, review the figure legend of figure 6 C-D, since the concentration of H4 indicated does not match the figure and does not indicate 2 different testing concentrations, besides PCR concentration is 8 µg/mL when in the following experiments is 80 µg/mL).

We are not sure why the lower concentration of H4 was not inhibited in this assay but the results were consistent. We appreciate the reviewer noticing the mistakes in the legend and these have been corrected.

6. In table 1 and 2, why higher concentrations (ng/mL) were tested for the mutant NCRD* than for SP-D. Is it the effect observed with the mutant NCR* due to the higher concentrations of protein there in comparison to SP-D (full length)?

We used higher concentrations of the NCRD trimers than for SP-D dodecamers due to the reduced neutralizing activity of the trimers. We wanted to use concentrations of the proteins that gave partial inhibition on their own to determine if there were additive effects when combined with H4. 

7. The reason behind the selection of the different concentrations of CRP, NCRD* and SP-D in the experiments in combination with H4 (results section B and C) should be, at a minimum, indicated or commented in the discussion. How the authors explain the different effects observed between SP-D and NCRD* in intracellular free calcium and MPO release?

Statements as “data not shown” should be avoided and the data/figure provided in supplemental material.

As noted above we used higher concentrations of NCRDs than SP-D to stay in the range of functional activity of these proteins for the various assays (e.g. the NCRDs require higher concentrations than SP-D to have effects). We removed the data not shown as noted above.

8. Are any of the peptides (CRP, H4, SP-D, NCRD and NCRD*) in an EDTA-containing buffer? In case they are, is the EDTA concentration being compensated with additional calcium before being added to the cells to exclude any EDTA-related effect in the experiments (for example in the MPO release, figure 10)?

These various proteins were not in EDTA buffer. This was done in the past for storage of SP-D but not more recently or in these experiments.

9. Recently, it has been shown that SP-D reduces LPS-induced NETosis in vivo and the detrimental effect of NETs in pulmonary surfactant (in mice lacking SP-D) (Arroyo, R., et al., 2020, Comm Biology), which should be referred in the discussion of SP-D ability to reduce NET mediated injury if authors want to discuss that idea.

We thank the reviewer for calling attention to this interesting article which we had missed and we now reference it in the Discussion.

---

## [Editor Report · Decision Letter 1]

10 Feb 2021

Histone H4 potentiates neutrophil inflammatory responses to influenza A virus: down-modulation by H4 binding to C-reactive protein and Surfactant protein D.

PONE-D-20-38304R1

Dear Dr. Hartshorn,

We’re pleased to inform you that your manuscript has been judged scientifically suitable for publication and will be formally accepted for publication once it meets all outstanding technical requirements.

Kind regards,

Nades Palaniyar, MSc., PhD.

Academic Editor

PLOS ONE

Additional Editor Comments (optional):

Reviewers' comments: The study and the revisions have been done, well.

---

## [Editor Report · Acceptance letter]

17 Feb 2021

PONE-D-20-38304R1 

Histone H4 potentiates neutrophil inflammatory responses to influenza A virus: down-modulation by H4 binding to C-reactive protein and Surfactant protein D. 

Dear Dr. Hartshorn:

I'm pleased to inform you that your manuscript has been deemed suitable for publication in PLOS ONE. Congratulations! Your manuscript is now with our production department. 

Kind regards, 

on behalf of

Dr. Nades Palaniyar 

Academic Editor

PLOS ONE